# GENERATIVE DISTRIBUTION DISTILLATION

## ABSTRACT

In this paper, we formulate the knowledge distillation (KD) as a conditional generative problem and propose the *Generative Distribution Distillation (GenDD)*. A naive *GenDD* encounters two major challenges: the curse of high-dimensional optimization and the lack of semantic supervision from labels. To address these issues, we introduce a *Split Tokenization (SplitTok)* strategy, achieving stable and effective unsupervised KD. Additionally, we develop the *Distribution Contraction* technique to integrate label supervision into the reconstruction objective. Our theoretical proof demonstrates that *GenDD* with *Distribution Contraction* serves as a gradient-level surrogate for multi-task learning, realizing efficient supervised training without explicit classification loss on multi-step sampling image representations. To evaluate the effectiveness of our method, we conduct experiments on balanced, imbalanced, and unlabeled data. Experimental results show that *GenDD* performs competitively in the unsupervised setting, significantly surpassing the KL baseline by **16.29%** on the ImageNet validation set. With label supervision, our ResNet-50 achieves **82.28%** top-1 accuracy on ImageNet in 600 epochs of training, establishing a new state-of-the-art. Code is available in the Appendix.

## 1 INTRODUCTION

For natural language tasks, both inputs and outputs reside in the same domain, *i.e.*, language sequences, enabling the unification of diverse tasks within a single generative model optimized via next-token prediction. ChatGPT and GPT4V (Ouyang et al., 2022; Achiam et al., 2023) exemplify this approach with data scaling law and are often regarded as an early prototype of artificial general intelligence (AGI), showcasing the effectiveness of generative learning in natural language. Motivated by this success, researchers have begun extending generative modeling to vision and multimodal domains (Liu et al., 2023; Li et al., 2024; Tian et al., 2024; Zhou et al., 2024; Fan et al., 2025; Wu et al., 2024; Yang et al., 2025), with the long-term goal of building AGI systems.

Two prominent classes of generative models have gained popularity in the vision domain: autoregressive models (Li et al., 2024; Tian et al., 2024) and diffusion models (Ho et al., 2020; Yang et al., 2025). Autoregressive models adopt the next-token prediction paradigm to sequentially generate image content, whereas diffusion models transform images into Gaussian noise through a forward diffusion process and learn to recover them via a reverse denoising process. In this paper, we recast knowledge distillation (KD) (Hinton et al., 2015), typically formulated as a discriminative task minimizing the KL divergence between categorical output distributions of the teacher and student, as a conditional generative problem modeled with the diffusion mechanism.

KD (Hinton et al., 2015) has been widely adopted for knowledge transfer and model compression in real-world applications. As illustrated in Figure 1, existing approaches typically guide the student model to imitate the teacher by minimizing either the KL divergence between output logits (Hinton et al., 2015; Zhao et al., 2022; Cui et al., 2024a; Lv et al., 2024) or the mean squared error (MSE) between intermediate-layer features (Chen et al., 2021; Romero et al., 2015; Park et al., 2019; Tian et al., 2020; Heo et al., 2019). These approaches introduce additional loss terms into a multi-task framework, increasing training complexity and requiring careful loss weight tuning. Figure 2 presents an empirical study on CIFAR-100, showing that student performance is highly sensitive to the choice of the loss weight. Moreover, the optimal weight also differs across teacher-student configurations, underscoring the limited robustness and generalizability of these methods in diverse application scenarios. This problem could exacerbate as the number of loss weight increases.

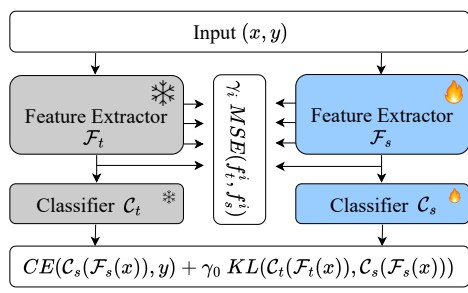

$$CE(\mathcal{C}_s(\mathcal{F}_s(x)), y) + \gamma_0 \, KL(\mathcal{C}_t(\mathcal{F}_t(x)), \mathcal{C}_s(\mathcal{F}_s(x)))$$

Figure 1: Previous methods are discriminative point-wise distillation.

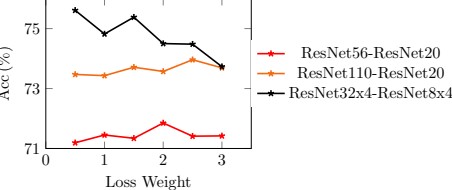

Figure 2: Sensitivity to loss weights of KD (Hinton et al., 2015). The accuracy of student models varies with different loss weights. Optimal loss weight varies with different teacher-student configurations.

**Generative Distribution Distillation (GenDD).** Inspired by the success of generative learning (Li et al., 2024; Ho et al., 2020; Song et al., 2020b), we leverage diffusion models to formulate KD as a single generative process. As shown in Figure 3, taking image representation $\mathcal{F}_s(x)$ of the student model as conditional inputs, we learn to generate the representation $\mathcal{F}_t(x)$ of the teacher model, thus achieving distribution mapping between the student and teacher.

**Challenges of GenDD.** MAR (Li et al., 2024) deploys a diffusion loss for autoregressive image generation. Specifically, images are tokenized in continuous token sequences via VAE (Kingma et al., 2013) and then fed into autoregressive models. However, image tokens from VAE only have a dimension of 16 while image representations in classification often have a large dimension, reaching to 2048. We empirically observe *the high-dimensional optimization disaster: the training can't converge or even crashes.* Moreover, diffusion models are optimized by variational lower bound (VLB) to reconstruct inputs, which *lacks semantic constraints with labels and thus hinders model performance*.

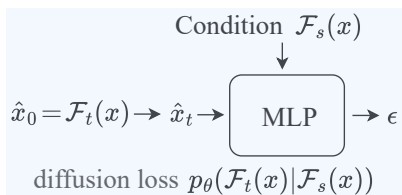

Figure 3: Conditional generation for KD.

**Our Solution.** To tackle the high-dimensional optimization challenge, we propose the *Split Tokenization (SplitTok)*: split image representation $\mathcal{F}_t(x)$ into token sequences with positional index. Conditoned on $\mathcal{F}_s(x)$, models are trained to reconstruct these tokens individually. Such a *SplitTok* operation effectively stabilizes the training of *GenDD*, achieving unsupervised KD. To enable label supervision of *GenDD*, we develop a *Distribution Contraction* technique. We theoretically prove that *GenDD* with *Distribution Contraction* serves as a surrogate to multi-task learning (combining reconstruction and classification loss), eliminating explicit classification loss and multi-step sampling and thus leading to efficient and effective supervised KD.

**Our Results.** To validate the effectiveness of our method, we conduct experiments on balanced data including CIFAR-100 (Krizhevsky & Hinton, 2009) and ImageNet (Russakovsky et al., 2015), imbalanced data like long-tailed ImageNet (Liu et al., 2019), and unlabeled data CC3M (Changpinyo et al., 2021). With the *SplitTok*, our *GenDD* model significantly surpasses the KL baseline by **16.29%** in the unsupervised KD setting. Moreover, with *Distribution Contraction technique*, *GenDD* incorporating label supervision largely outperforms previous distillation methods. Especially, we achieve a new state-of-the-art ResNet-50 performance on ImageNet.

In summary, our contributions are as follows:

- We formulate KD as a conditional generation problem and propose the *GenDD* algorithm.

- To address the high-dimensional optimization challenge, we propose a *SplitTok* strategy. To enable label supervision of *GenDD*, the *Distribution Contraction* technique is developed.

- We theoretically prove that *GenDD* with *Distribution Contraction* acts as a gradient-level surrogate for the multi-task learning, resulting in effecient and effective optimization.

- We empirically show the advantage of our method on balanced, imbalanced, and unlabeled data. Specifically, we achieve the state-of-the-art performance on ImageNet for both unsupervised and supervised KD settings.

Table 1: Comparison with previous distillation methods.

| Method | Generative or Discriminative | Distributional or Point-wise | Sensitivity to Loss Weight |
|--------|------------------------------|------------------------------|----------------------------|
| Logits-based | discriminative | point-wise | sensitive |
| Feature-based | discriminative | point-wise | sensitive |
| **GenDD** | generative | distributional | NA |

## 2 RELATED WORK

**Generative Learning.** In NLP, generative modeling based on next-token prediction (NTP) forms the foundation of the GPT series of language models (Radford et al., 2018; 2019; Brown et al., 2020; Ouyang et al., 2022; Achiam et al., 2023). This approach enables unsupervised learning from large-scale corpora and has driven significant advances in zero-shot and few-shot generalization, culminating in powerful systems such as ChatGPT and GPT4V (Ouyang et al., 2022; Achiam et al., 2023) that exhibit strong performance across diverse natural language tasks without task-specific fine-tuning. Subsequently, autoregressive (AR) models leveraging NTP have also gained popularity in vision (Li et al., 2024; Tian et al., 2024) and multi-modal (Liu et al., 2023; Zhu et al., 2023) domains, fostering the development of unified generalist models capable of both understanding and generation (Zhou et al., 2024; Fan et al., 2025; Wu et al., 2024; Yang et al., 2025) on multi-modal data.

Besides NTP and autoregressive (AR) models, diffusion models (Sohl-Dickstein et al., 2015; Song & Ermon, 2019; Ho et al., 2020; Song et al., 2020b) have emerged as a powerful class of generative methods, demonstrating impressive sample quality and robustness (Rombach et al., 2022; Yang et al., 2025; Zhou et al., 2024). However, they often require multi-step iterative sampling during inference, which can be computationally expensive and time-consuming. Recently, flow matching approaches (Geng et al., 2025; Lipman et al., 2022; Gat et al., 2024) have been proposed to address these limitations by providing efficient and scalable generative modeling with fewer sampling steps while maintaining high fidelity.

**Knowledge Distillation.** Knowledge distillation (KD) (Hinton et al., 2015) is developed to transfer "dark knowledge" from a teacher model to a student model. The core idea is to leverage the soft targets, *i.e.*, the probability distribution over classes, produced by the teacher to guide the training of the student. These soft labels contain rich information about inter-class similarities that are not captured by one-hot labels, thereby enabling the student to learn more generalizable representations. Since the success of KD (Hinton et al., 2015), advanced logit-based (Furlanello et al., 2018; Zhang et al., 2018; Cho & Hariharan, 2019; Huang et al., 2022; Zhao et al., 2022; Hao et al., 2023; Cui et al., 2024a; 2025) and feature-based (Romero et al., 2015; Park et al., 2019; Tian et al., 2020; Heo et al., 2019; Chen et al., 2021; Huang et al., 2023) algorithms have been proposed. However, these complicated methods are often sensitive to loss weights and require hyperparameter tuning for each teacher-student configuration.

Additionally, previous KD methods are typically trained along with discriminative cross-entropy loss and promote consistency between the teacher and student on each data point. In this paper, the proposed *GenDD* is optimized with a single reconstruction objective and models the distribution of each example. Refer to Table 1 for the comparison summary with previous work.

## 3 METHOD

### 3.1 REVISITING KNOWLEDGE DISTILLATION AS MULTI-TASK LEARNING

A typical classification model comprises a feature extractor $\mathcal{F}(\cdot)$ and a classifier $\mathcal{C}(\cdot)$. Given an input image $x$, the model produces a feature representation $\mathcal{F}(x)$ and a corresponding prediction $\arg\max \mathcal{C}(\mathcal{F}(x))$. While knowledge distillation (KD) is broadly applicable to a wide range of real-world scenarios, we focus on image classification in this work. KD is designed to transfer the inductive knowledge of the teacher model to the student model, enabling both model compression and improved generalization. Previous KD methods (Hinton et al., 2015; Tian et al., 2020; Chen et al., 2021) are often discriminative and point-wise, minimizing KL divergence or Mean Square

Error (MSE) between sample logits or intermediate-layer features,

$$\min_{\theta} CE(\mathcal{C}_s \circ \mathcal{F}_s(x), y) + \gamma_0 \cdot KL(\mathcal{C}_t \circ \mathcal{F}_t(x), \mathcal{C}_s \circ \mathcal{F}_s(x)) + \sum_{i=1} \gamma_i \cdot MSE(f_t^i, f_s^i), \quad (1)$$

where $\theta$ is parameters of student, $\{\gamma_i\}$ are hyper-parameters for multi-task learning.

As illustrated in Figure 2, the performance of the student model is notably sensitive to the choice of hyperparameters, even on the *same training dataset* with different teacher-student configurations, making the optimization challenging. For more detailed comparisons with advanced KD methods, please refer to Appendix A.2. Moreover, in real-world scenarios, the teacher can be trained on custom data that can't be accessed publicly because of *privacy protection*. In this case, previous algorithms can't work well without cross-entropy in Equation (1), which is validated in Section 4.1.

In contrast, our proposed *Generative Distribution Distillation (GenDD)* is optimized by a unified *reconstruction objective*, eliminating the need for extensive hyperparameter tuning. Furthermore, it achieves competitive performance using only unlabeled data (Section 3.2.1) and attains state-of-the-art results when annotation supervision is available (Section 3.2.2).

## 3.2 GENERATIVE DISTRIBUTION DISTILLATION (GENDD)

Inspired by the success of ChatGPT (Ouyang et al., 2022; Achiam et al., 2023) in natural language processing (NLP), recent efforts have aimed to unify multi-modal understanding and generation within a single generative framework (Zhou et al., 2024; Wu et al., 2024; Yang et al., 2025; Fan et al., 2025). In particular, diffusion models have recently emerged as promising alternatives to large language models (LLMs) (Arriola et al., 2025; Yang et al., 2025). In this paper, we propose to formulate KD as a generative learning process based on diffusion foundations.

### 3.2.1 GENDD WITH UNLABELED DATA

Given an image $x \in X$, $\hat{x}_0 \sim q(\mathcal{F}_t(x))$, taking $\mathcal{F}_s(x)$ as condition, we learn to reconstruct the image representation of teacher model, *i.e.*, $\hat{x}_0$, with the following training objective,

$$\mathbb{E}_{x,m,\epsilon}\left[||\epsilon - \epsilon_{\theta}(\hat{x}_m, m, \mathcal{F}_s(x))||^2\right], \quad (2)$$

where $\epsilon \in \mathcal{N}(\mathbf{0}, \mathbf{I})$, $m \in [0, M]$ is the sampled time step ($M$ is the maximum), $\hat{x}_m = \sqrt{\bar{\alpha}_m}\hat{x}_0 + \sqrt{1 - \bar{\alpha}_m}\epsilon$ is the noisy input at time step $m$, in particular, $\hat{x}_0 = \mathcal{F}_t(x)$, $\bar{\alpha}_m = \Pi_{i=1}^{m}\alpha_i$, and $\alpha$ is defined with a variance schedule (Ho et al., 2020; Nichol & Dhariwal, 2021).

At inference, with an input image $x$ and a sampled $\hat{x}_M' \in \mathcal{N}(\mathbf{0}, \mathbf{I})$, the image representation $\hat{x}_0'$ could be generated through iterative update from $m = M$ to $m = 0$:

$$\hat{x}_{m-1}' = \frac{1}{\sqrt{\alpha_m}}\left(\hat{x}_m' - \frac{1 - \alpha_m}{\sqrt{1 - \bar{\alpha}_m}}\epsilon_{\theta}(\hat{x}_m', m, \mathcal{F}_s(x))\right) + \sigma_m\epsilon, \quad (3)$$

where $\epsilon \in \mathcal{N}(\mathbf{0}, \mathbf{I})$, $\sigma_m$ could be learned or pre-defined. Note that, the reverse diffusion process could be respaced (Li et al., 2024; Song et al., 2020a) for efficient sampling.

Then, the final prediction could be derived by inputting the image representation $\hat{x}_0'$ into teacher model classifier, *i.e.*, $\arg\max \mathcal{C}_t(\hat{x}_0')$.

**High-dimensional Optimization Disaster.** Following MAR (Li et al., 2024), we implement the diffusion head using a 3-layer MLP. MAR (Li et al., 2024) showcases the effectiveness of continuous tokenizers for autoregressive image generation, where images are first tokenized into sequences of continuous tokens using a VAE (Kingma et al., 2013). These tokens are then fed into an autoregressive model that learns the per-token distribution through a diffusion loss.

However, the dimensionality of each token in VAE (Kingma et al., 2013) is limited to 16, whereas feature representations in image classification tasks typically have much higher dimensionality, reaching up to 2048. Our empirical study reveals a high-dimensional optimization issue, particularly when the feature dimension of the student model, $\mathrm{Dim}(\mathcal{F}_s(x))$, is much lower than that of the teacher model, $\mathrm{Dim}(\mathcal{F}_t(x))$, often leading to training instability or failure. Refer to Section 4.3 for more details.

**Split Tokenization (SplitTok).** To address the challenges of high-dimensional optimization, we propose decomposing the feature representation $\hat{x}_0$ into a sequence of non-overlapped lower-dimensional tokens. Specifically, we define the *SplitTok* operation as:

$$SplitTok(\mathcal{F}_t(x)) = \left[(\hat{x}_0^1, 1, \mathcal{F}_s(x)), (\hat{x}_0^2, 2, \mathcal{F}_s(x)), \dots, (\hat{x}_0^n, n, \mathcal{F}_s(x))\right], \tag{4}$$

where each tuple consists of a token $\hat{x}_0^i$, its position index $i$, and the conditioning context from the student model $\mathcal{F}_s(x)$. Based on this structure, we reformulate the training objective in Equation (2) into a token-wise form:

$$\mathbb{E}_{y,\mathrm{id},c,m,\epsilon}\left[||\epsilon - \epsilon_\theta(\hat{y}_m, m, \mathrm{id}, c)||^2\right], \quad \text{where } (y, \mathrm{id}, c) \sim q\left(SplitTok(\mathcal{F}_t(x))[\mathrm{id}]\right). \tag{5}$$

This token-based formulation allows the model to operate in lower-dimensional subspaces, thereby mitigating instability during optimization in high-dimensional feature spaces.

### 3.2.2 GenDD with Label Supervision

Conditioned on the student's feature representation, *GenDD* reconstructs the teacher's feature tokens, enabling unsupervised knowledge distillation. However, the reconstruction objective alone fails to exploit label supervision during training. To address this problem, we introduce a *Distribution Contraction* mechanism that enables *GenDD* to effectively incorporate label information into the optimization process.

**Multi-task Learning.** To incorporate label supervision, a straightforward baseline is multi-task learning, which combines the reconstruction objective with a standard cross-entropy loss:

$$\min_{\theta_s} \mathcal{L}_{\mathrm{CE}} = -y \log \mathcal{C}_s(\hat{x}_0'),$$
$$\text{s.t.} \min_\theta \mathbb{E}_{y,\mathrm{id},c,m,\epsilon}\left[\left|\epsilon - \epsilon_\theta(\hat{y}_m, m, \mathrm{id}, c)\right|^2\right], \tag{6}$$

where $\mathcal{C}_s$ denotes the classifier on top of the reconstructed representation $\hat{x}_0'$, $\theta = (\theta_s, \theta_{diff})$ are parameters of student model and diffusion head respectively.

As shown in Equation (6), the cross-entropy loss encourages the generated feature $\hat{x}_0'$ to be correctly classified, while the reconstruction loss regularizes the student feature space to align with that of the teacher. These two objectives can be optimized either alternately or simultaneously during training.

However, in practice, involving $\hat{x}_0'$ directly in the training of the diffusion model is computationally inefficient. Since $\hat{x}_0'$ must be sampled through a multi-step reverse diffusion process at each iteration, using it as an intermediate target for supervision substantially increases training time and resource consumption. Moreover, gradients cannot be efficiently propagated through the sampling chain, limiting the effectiveness of end-to-end optimization.

**GenDD with Label Supervision.** Instead of relying on conventional multi-task learning, we incorporate label supervision through the proposed *Distribution Contraction* technique, formally defined in Definition 1. Furthermore, Theorem 1 establishes that *GenDD*, trained with the *Distribution Contraction* technique, serves as an efficient and effective surrogate for multi-task learning.

**Definition 1 (Distribution Contraction)** Let $\hat{x}_0 \sim q(\mathcal{F}_t(x))$ be the feature representation produced by a well-trained teacher model $\mathcal{F}_t$ on input $x \in X$, where $\mathcal{C}_t(\hat{x}_0) \in \mathbb{R}^C$ is the logits vector over $C$ categories. To incorporate label supervision $y$ during diffusion model training, we enhance the semantic consistency of $\hat{x}_0$ by contracting it toward the class center $c_y$:

$$\tilde{x}_0 = \lambda\hat{x}_0 + (1 - \lambda)c_y, \tag{7}$$

where $\lambda \in [0, 1]$ controls the degree of contraction, $c_y$ denotes the centroid of features for class $y$.

**Theorem 1 (Distribution Contraction Approximates Multi-task Learning at Gradient Level)**
*Assume the teacher model, composed of a feature extractor $\mathcal{F}_t(\cdot)$ and a linear classifier $\mathcal{C}_t(\cdot)$, is well-trained, then the optimization of the reconstruction objective with distribution contraction in Definition 1:*

$$\mathcal{L}_{GenDD} = \mathbb{E}_{x,m,\epsilon}\left[\|\epsilon - \epsilon_\theta(\tilde{x}_m, m, \mathcal{F}_s(x))\|^2\right], \tag{8}$$

*is approximately equivalent, at the gradient level, to optimizing the multi-task objective:*

$$\mathcal{L}_{multi} = \gamma_0 \, \mathbb{E}_{x,m,\epsilon} \left[ \| \epsilon - \epsilon_\theta(\hat{x}_m, m, \mathcal{F}_s(x)) \|^2 \right] + \gamma_1 \, \mathbb{E}_x \left[ \mathcal{L}_{\text{CE}} \left( \mathcal{C}_t(\hat{x}_0'), y \right) \right], \quad (9)$$

*where $\gamma_0$ and $\gamma_1$ are constants controlling the relative weights of the two loss terms, $\mathcal{C}_t(\cdot)$ is frozen.*

*Proof. See Appendix A.1.*

**Remark.** Theorem 1 shows that $\mathcal{L}_{\text{GenDD}}$ acts as a gradient-level surrogate for the multi-task objective $\mathcal{L}_{\text{multi}}$, avoiding explicit classification loss optimization and eliminating the need for multi-step sampling to obtain $\hat{x}_0'$ during training. This enables more efficient training while retaining strong performance with label supervision.

# 4    EXPERIMENTS

Section 4.1 presents the competitive performance of *GenDD* under the unsupervised KD setting. When label supervision is incorporated via *Distribution Contraction*, *GenDD* achieves strong results on both balanced and imbalanced datasets, as shown in Section 4.2. Finally, we perform ablation studies in Section 4.3 to assess the impact of the proposed *Split Tokenization (SplitTok)* and *Distribution Contraction* techniques.

**Experimental Settings.** Following prior work (Li et al., 2024), we implement the diffusion head using a 3-layer MLP. For training, the maximum diffusion step is set to $M = 1000$. At inference, we apply a 64-step sampling procedure to generate the feature representation $\hat{x}_0'$. For *SplitTok*, the feature representation $\hat{x}_0$ is divided into non-overlapping tokens, each with a dimensionality of 64. To enhance generation quality, we employ classifier-free guidance with a scale of 2.0.

For the unsupervised KD setting, we evaluate *GenDD* on the target dataset, *i.e.*, ImageNet (Deng et al., 2009), and non-target dataset CC3M (Changpinyo et al., 2021). Under the supervised KD setting, we train various teacher-student configurations on balanced (including ImageNet (Deng et al., 2009) and CIFAR (Krizhevsky et al., 2009)) and imbalanced data (ImageNet-LT).

## 4.1    GENDD IN UNSUPERVISED SETTING

To evaluate the effectiveness of *GenDD* for unsupervised knowledge distillation (KD), we train models on both target data (ImageNet) and non-target data (CC3M), and assess their performance on the ImageNet validation set. The results are summarized in Table 3.

For teacher-student configurations such as (ResNet-34, ResNet-18) and (ResNet-50, MobileNet), we adopt pre-trained teacher models from PyTorch. Since these teachers have been trained on the target dataset, their predictions closely approximate the ground-truth, allowing conventional KL-based distillation without cross-entropy to perform competitively relative to *GenDD*.

However, in practical unsupervised KD scenarios, custom training data, along with their annotations, can be both private and inaccessible. To simulate this setting, we train student models on non-target data, specifically CC3M (Changpinyo et al., 2021), where teacher models have never been exposed to the data. In this case, teacher predictions become less reliable, and naive KL-based distillation without cross-entropy for label supervision fails to produce satisfactory results. As shown in Table 3, with *GenDD*, our MobileNet achieves 67.89 top-1 accuracy, significantly outperforms the KL baseline by **16.29%**.

## 4.2    GENDD WITH LABEL SUPERVISION

Our studies on balanced data, including CIFAR and ImageNet, are presented in Section 4.2.1. Section 4.2.2 discusses the effects of *GenDD* on imbalanced data, *i.e.*, ImageNet-LT (Liu et al., 2019).

### 4.2.1    EXPERIMENTAL RESULTS ON BALANCED DATA

**Experimental Results on ImageNet.** On ImageNet, we evaluate a range of teacher-student configurations, covering diverse network architectures (CNNs with regular or depth-wise convolutions, and

Table 2: **Top-1 accuracy (%) on the ImageNet validation with supervised GenDD**. All results are the average over three trials. "*" represents that the models are reproduced with the cosine learning rate schedule for fair comparison.

| | Teacher | Student | Discriminative Point-wise Distillation | | | | | | | Gen.D.D. |
| | | | AT | OFD | CRD | ReviewKD | DKD* | IKL-KD* | KD* | GenDD |
|---|---|---|---|---|---|---|---|---|---|---|
| | | | *ResNet-34, ResNet-18, Regular recipe, 100 epochs* | | | | | | | |
| Top-1 | 73.31 | 69.75 | 70.69 | 70.81 | 71.17 | 71.61 | 71.87 | 71.91 | 71.24 | **72.38** |
| Top-5 | 91.42 | 89.07 | 90.01 | 89.98 | 90.13 | 90.51 | 90.45 | 90.52 | 90.23 | **90.63** |
| | | | *ResNet-50, MobileNet, Regular recipe, 100 epochs* | | | | | | | |
| Top-1 | 76.16 | 68.87 | 69.56 | 71.25 | 71.37 | 72.56 | 72.55 | 73.19 | 71.44 | **73.78** |
| Top-5 | 92.86 | 88.76 | 89.33 | 90.34 | 90.41 | 91.00 | 91.05 | 91.47 | 90.35 | **91.56** |
| | | | *BEiTv2, ResNet-50, Strong recipe, 300 (A2) or 600 (A1) epochs* | | | | | | | |
| Top-1 (BEiT-L-A2) | 88.01 | 79.80 | - | - | 79.48 | 79.11 | 80.77 | 80.98 | 80.89 | **81.64** |
| Top-1 (BEiT-B-A2) | 86.12 | 79.80 | - | - | - | - | - | - | 80.96 | **81.76** |
| Top-1 (BEiT-L-A1) | 88.01 | 80.38 | - | - | - | - | 81.83 | - | 81.68 | **82.28** |

Table 3: **Top-1 accuracy(%) on the ImageNet validation with unsupervised GenDD.**

| Method | Teacher | Student | Accuracy |
|---|---|---|---|
| w/o Label On Target Data, *i.e.*, ImageNet-1K | | | |
| KL | ResNet-50 | MobileNet | 71.40 |
| **GenDD** | ResNet-50 | MobileNet | **72.03** |
| w/o Label On Non-target Data, *i.e.*, CC3M | | | |
| KL | ResNet-50 | MobileNet | 51.60 |
| **GenDD** | ResNet-34 | ResNet-18 | **66.90** |
| **GenDD** | ResNet-50 | MobileNet | **67.89** |

Table 4: **Top-1 accuracy(%) on the ImageNet-LT validation with GenDD.** "*" represents the unsupervised setting.

| Method | Teacher | Student | Accuracy |
|---|---|---|---|
| Baseline | - | ResNet-18 | 41.15 |
| Baseline | - | ResNet-50 | 45.47 |
| KD | ResNeXt-101 | ResNet-18 | 44.32 |
| KD | ResNeXt-101 | ResNet-50 | 48.31 |
| IKL-KD | ResNeXt-101 | ResNet-18 | 45.21 |
| IKL-KD | ResNeXt-101 | ResNet-50 | 49.29 |
| **GenDD*** | ResNext-101 | ResNet-18 | **45.54** |
| **GenDD*** | ResNeXt-101 | ResNet-50 | **49.31** |

Table 5: **Top-1 accuracy (%) on the CIFAR-100 validation.** Teachers and students are in the **same** architectures. $\Delta$ represents the improvements over the KD (Hinton et al., 2015) baseline. All results are the average over three trials.

| Distillation Manner | Teacher | ResNet56 72.34 | ResNet110 74.31 | ResNet32×4 79.42 | WRN-40-2 75.61 | WRN-40-2 75.61 | VGG13 74.64 |
| | Student | ResNet20 69.06 | ResNet32 71.14 | ResNet8×4 72.50 | WRN-16-2 73.26 | WRN-40-1 71.98 | VGG8 70.36 |
|---|---|---|---|---|---|---|---|
| Discriminative Point-wise Distillation | FitNet | 69.21 | 71.06 | 73.50 | 73.58 | 72.24 | 71.02 |
| | RKD | 69.61 | 71.82 | 71.90 | 73.35 | 72.22 | 71.48 |
| | CRD | 71.16 | 73.48 | 75.51 | 75.48 | 74.14 | 73.94 |
| | OFD | 70.98 | 73.23 | 74.95 | 75.24 | 74.33 | 73.95 |
| | ReviewKD | 71.89 | 73.89 | 75.63 | 76.12 | 75.09 | 74.84 |
| | DKD | 71.97 | 74.11 | 76.32 | 76.24 | 74.81 | 74.68 |
| | IKL-KD | 71.44 | 74.26 | 76.59 | 76.45 | 74.98 | **74.98** |
| | KD | 70.66 | 73.08 | 73.33 | 74.92 | 73.54 | 72.98 |
| Gen.D.D. | **GenDD** | **72.63** | **74.95** | **77.47** | **76.83** | **75.98** | 74.24 |
| | $\Delta$ | **+1.97** | **+1.87** | **+4.14** | **+1.91** | **+2.44** | **+1.26** |

Transformers), training recipes (standard vs. strong augmentation), and model scales (e.g., ResNet-34, ResNet-50, BEiT-Large). *Despite the significant variation across configurations, we employ a consistent* $\lambda = 0.9$ *for the Distribution Contraction in Definition 1, highlighting the generalizability, robustness, and practical convenience of GenDD.*

Under the regular training recipe (including *RandomResizedCrop* and horizontal flip), we train models 100 epochs with a cosine learning rate schedule. For fair comparisons, we reproduce the results of KD, DKD, and IKL-KD with their open-sourced code and just replace the step learning rate schedule with the cosine learning rate schedule. Our ResNet-18 achieves a top-1 accuracy of 72.38%, outperforming KD, IKL-KD, and DKD by 1.14%, 0.47%, and 0.51%, respectively. Similarly, our MobileNet reaches 73.78% top-1 accuracy, surpassing KD, IKL-KD, and DKD by 2.34%, 0.59%, and 1.23%, respectively.

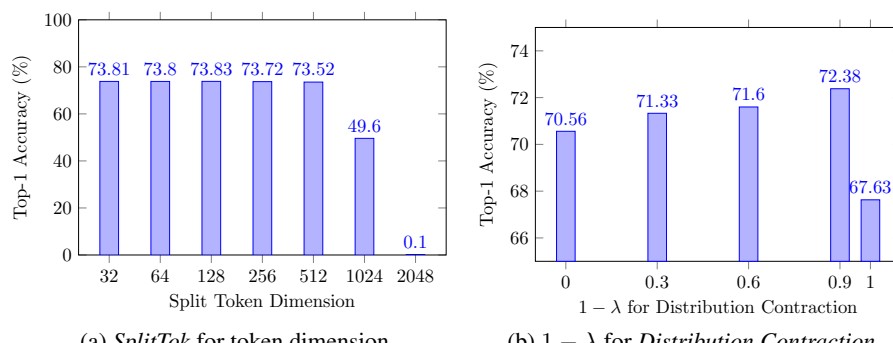

(a) *SplitTok* for token dimension  (b) $1 - \lambda$ for *Distribution Contraction*

Figure 4: **Ablation studies on *SplitTok* and *Distribution Contraction*.** (a) Top-1 Accuracy with different token dimension for *SplitTok*. The teacher-student configuration of (ResNet-50, MobileNet) is used on ImageNet.; (b) Top-1 Accuracy under different $1 - \lambda$ values for *Distribution Contraction*. The teacher-student configuration of (ResNet-34, ResNet-18) is used on ImageNet.

When applying a strong training recipe, prior work (Hao et al., 2023) shows that recent advanced KD methods such as DKD and ReviewKD only perform comparably to the original KD:

- A1: *RandAug(7/0.5)*, *MixUp*: 0.2, *CutMix*: 1.0, *Label Smoothing*: 0.1, training 600 epochs.
- A2: *RandAug(7/0.5)*, *MixUp*: 0.1, *CutMix*: 1.0, *Label Smoothing*: 0.0, training 300 epochs.

Remarkably, our *GenDD* models consistently outperform these baselines by a significant margin with the same training settings. Specifically, taking BEiTv2-Large as the teacher, our ResNet-50 achieves 82.28% top-1 accuracy with the A1 training recipe.

**Experimental Results on CIFAR.** Following previous work (Cui et al., 2024a; Chen et al., 2021), we consider the distillation among the architectures having the same unit structures, like ResNet56 and ResNet20, VGGNet13 and VGGNet8. On the other hand, we also explore the distillation among architectures made up of different unit structures, like WideResNet and ShuffleNet, VGGNet and ResNet. Specifically, we train all models for 240 epochs with a learning rate that decays by 0.1 at the 150th, 180th, and 210th epoch.

Experimental results on CIFAR-100 are summarized in Table 5 and Table 7 (Appendix). Table 5 lists the comparisons with previous methods under the setting that the architectures of the teacher and student have the same unit structures. As shown in Table 5, *GenDD* models can achieve much better or comparable performance in all considered settings. Specifically, we achieve the best performance in 5 out of 6 training settings. Table 7 lists the comparisons with previous methods under the setting that the architectures of the teacher and student have different unit structures. As shown in Table 7, we achieve the best performance in 4 out of 5 training configurations.

### 4.2.2 EXPERIMENTAL RESULTS ON IMBALANCED DATA

Real-world data often exhibits a long-tailed distribution, making long-tailed recognition a critical challenge for practical applications. Extensive research has been devoted to addressing this problem through both algorithmic and theoretical advances (Cui et al., 2019; Cao et al., 2019; Kang et al., 2019; Cui et al., 2022; Menon et al., 2020; Cui et al., 2021; 2023; 2024b). Following recent efforts (Cui et al., 2024a; 2025), we also evaluate the effectiveness of *GenDD* under data imbalance using the ImageNet-LT benchmark (Liu et al., 2019). We train ResNet models for 90 epochs using *RandomResizedCrop* and horizontal flipping as standard preprocessing. Following previous work (Cui et al., 2024a), we report top-1 accuracy across Many-shot, Medium-shot, Few-shot, and All classes to comprehensively assess performance.

As shown in Table 4 and Table 9 (Appendix A.5), we observe an interesting phenomenon: *GenDD without label supervision can even achieve slightly better performance than IKL-KD incorporating labels. However, there are few accuracy gains after applying the label supervision with Distribution Contraction, which is a different behaviour compared to balanced data.* This phenomenon gives us new insight into KD for imbalanced data: the necessity of labels for KD on imbalanced data. As this work focuses on generative learning of KD, we leave this problem as our future work.

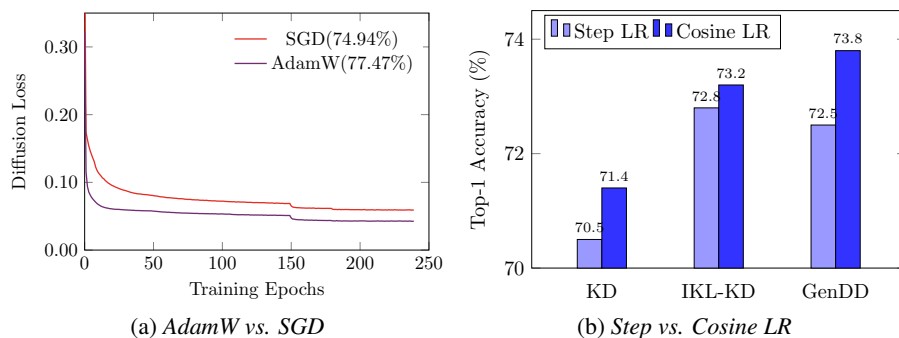

(a) *AdamW vs. SGD*        (b) *Step vs. Cosine LR*

Figure 5: **Ablation studies on optimizer and learning rate schedule.** (a) Comparison between AdamW and SGD optimizer for GenDD with teacher-student configuration of (ResNet-32x4, ResNet-8x4) on CIFAR; (b) Comparison between Step and Cosine learning rate schedule for GenDD with teacher-student configuration of (ResNet50, MobileNet) on ImageNet.

### 4.3 ABLATION STUDIES

**Ablation of *SplitTok*.** We validate the necessity of the proposed *SplitTok* under the (ResNet-50, MobileNet) teacher-student configuration on ImageNet. As shown in Figure 4a, the model maintains competitive performance when the token dimension is $\leq 256$. However, accuracy drops sharply to $0.1\%$ as the token dimension increases from $512$ to $2048$, highlighting the high-dimensional optimization challenge and the effectiveness of *SplitTok* in mitigating it.

**Ablation on $\lambda$ for *Distribution Contraction*.** We validate the effectiveness of the proposed *Distribution Contraction* technique under the (ResNet-34, ResNet-18) teacher configuration on ImageNet. As illustrated in Figure 4b, *GenDD* achieves competitive performance in the unsupervised KD setting with $1 - \lambda = 0.0$. Deploying *Distribution Contraction* technique with a proper $1 - \lambda = 0.9$, our model achieves significant performance gains. Interestingly, we observe that the model achieves much worse accuracy when the sample features contract to class centers, which indicates the importance of the continuity of the sample feature space. Inspired by this phenomenon, we apply the unsupervised mixup for diffusion training.

**AdamW vs. SGD Optimizer.** We investigate the impact of different optimizers on training *GenDD*. While SGD is commonly used for CNNs and AdamW/Adam are standard choices for Transformers, we adopt AdamW for *GenDD* following previous work MAR (Liu et al., 2023) and DDPM (Ho et al., 2020). As shown in Figure 5a, our empirical results demonstrate that AdamW leads to more stable and effective optimization for *GenDD*.

**Cosine vs. Step Learning Rate Schedule for GenDD.** We study the impact of learning rate schedules on *GenDD*, focusing on step decay and cosine annealing strategies. For fair comparison, we adopt the step schedule on CIFAR and reproduce the results of previous work with a cosine learning rate in their open-sourced code on ImageNet. Our empirical results show that the cosine schedule is critical for *GenDD*, particularly on large-scale datasets such as ImageNet. As illustrated in Figure 5b, cosine learning rates significantly accelerate convergence and improve overall performance.

## 5 CONCLUSION, LIMITATION, AND FUTURE WORK

In this paper, we propose the *Generative Distribution Distillation (GenDD)* algorithm, formulating the KD as a conditional generation problem. The straightforward *GenDD* pipeline suffers from the high-dimensional optimization disaster and the lack of label supervision. We propose the *Split Tokenization (SplitTok)* and *Distribution Contraction* techniques to address the above issues, respectively. With theoretical analysis, we prove that *GenDD* with *Distribution Contraction* approximates the multi-task learning (combining the reconstruction loss and the cross-entropy loss), while eliminating the multi-step sampling during training and achieving efficient optimization. Experimental results in unsupervised/supervised KD demonstrate the effectiveness of our method.

At inference, we adopt a 64-step sampling to generate image representations for classification, which can cause slightly higher latency. We would explore the few-step diffusion models or flow matching to overcome this limitation in future work.

# 6 ETHICS STATEMENT

This work presents the GenDD framework, aiming at improving the efficiency and effectiveness of training smaller student models. By enabling compact models to better approximate the performance of larger teacher models, our method has the potential to reduce computational costs and energy consumption, thereby contributing to more sustainable and accessible AI.

We acknowledge that advances in KD may also be misused, for example, to replicate proprietary models without authorization or to reduce safeguards embedded in larger models. To mitigate such risks, we emphasize that our work is released for academic research purposes. All experiments are conducted on standard public datasets (CIFAR, ImageNet, CC3M), and no sensitive or personally identifiable information is used.

# 7 REPRODUCIBILITY STATEMENT

To ensure the reproducibility of our research, we provide detailed information regarding our experimental setup. All hyperparameters and implementation specifics are thoroughly documented in Section 4 and Table A.2 in the Appendix of this paper. Additionally, our proof of Theorem 1 is provided in Section A.1 of the Appendix. Finally, the code can be accessed at the following URL:

```
https://drive.google.com/file/d/12bwEj-wUqy2LFGpwsc_bikZF-Y_5SaP8/
view?usp=sharing
```

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

# A APPENDIX

## A.1 PROOF OF THEOREM 1

We begin by restating the *GenDD* objective and the *multi-task* objective. The *GenDD* objective is the expected noise prediction loss over noisy data $\tilde{x}_m$:

$$\mathcal{L}_{\text{GenDD}} = \mathbb{E}_{x,m,\epsilon} \left[ \|\epsilon - \epsilon_\theta(\tilde{x}_m, m, \mathcal{F}_s(x))\|^2 \right], \tag{10}$$

where $\tilde{x}_m$ is given by:

$$\tilde{x}_m = \sqrt{\bar{\alpha}_m}\tilde{x}_0 + \sqrt{1 - \bar{\alpha}_m}\epsilon, \quad \tilde{x}_0 = \lambda\hat{x}_0 + (1 - \lambda)c_y \text{ (by Definition 1)}, \tag{11}$$

where $\hat{x}_0$ is the teacher feature and $c_y$ is the class center for class $y$.

The multi-task objective consists of two parts: - The reconstruction loss via noise prediction:

$$\mathcal{L}_{\text{noise}} = \mathbb{E}_{x,m,\epsilon} \left[ \|\epsilon - \epsilon_\theta(\hat{x}_m, m, \mathcal{F}_s(x))\|^2 \right],$$

- The classification loss via cross-entropy:

$$\mathcal{L}_{\text{CE}} = \mathbb{E}_x \left[ \mathcal{L}_{\text{CE}} \left( \mathcal{C}_t(\hat{x}_0'), y \right) \right],$$

where $\hat{x}_m$ is the noisy version of $\hat{x}_0$, $\mathcal{C}_t$ is the teacher classifier, and $\hat{x}_0'$ is the generated feature.

Thus, the multi-task loss is:

$$\mathcal{L}_{\text{multi}} = \gamma_0 \mathcal{L}_{\text{noise}} + \gamma_1 \mathcal{L}_{\text{CE}},$$

where $\gamma_0$ and $\gamma_1$ are scaling constants.

**Gradients of $\mathcal{L}_{GenDD}$ Regarding $\hat{x}_0'$.**

With the single-step estimation of $\hat{x}_0'$,

$$\hat{x}_0' = \frac{1}{\sqrt{\bar{\alpha}_m}}(\tilde{x}_m - \sqrt{1 - \bar{\alpha}_m}\epsilon_\theta). \tag{12}$$

Then substitute $\tilde{x}_m$ in Equation (11),

$$\hat{x}_0' = \frac{1}{\sqrt{\bar{\alpha}_m}}(\sqrt{\bar{\alpha}_m}\tilde{x}_0 + \sqrt{1 - \bar{\alpha}_m}\epsilon - \sqrt{1 - \bar{\alpha}_m}\epsilon_\theta) = \tilde{x}_0 + \frac{\sqrt{1 - \bar{\alpha}_m}}{\sqrt{\bar{\alpha}_m}}(\epsilon - \epsilon_\theta), \tag{13}$$

$$\epsilon - \epsilon_\theta = \sqrt{\frac{\bar{\alpha}_m}{1 - \bar{\alpha}_m}}(\hat{x}_0' - \tilde{x}_0) \tag{14}$$

With Equation (12),

$$\mathcal{L}_{GenDD} = \mathbb{E}_{x,m,\epsilon} \left[ \|\epsilon - \epsilon_\theta(\tilde{x}_m, m, \mathcal{F}_s(x))\|^2 \right] \tag{15}$$

$$= \mathbb{E}_{x,m,\epsilon} \left[ \left\| \epsilon - \frac{1}{\sqrt{1 - \bar{\alpha}_m}}(\tilde{x}_m - \sqrt{\bar{\alpha}_m}\hat{x}_0') \right\|^2 \right] \tag{16}$$

$$= \mathbb{E}_{x,m,\epsilon} \left[ \|\epsilon - \epsilon_\theta(\hat{x}_0')\|^2 \right]. \tag{17}$$

We take gradients of $\mathcal{L}_{GenDD}$ with respect to $\hat{x}_0'$,

$$\nabla_{\hat{x}_0'} \mathcal{L}_{GenDD} = 2\sqrt{\frac{\bar{\alpha}_m}{1 - \bar{\alpha}_m}}(\epsilon - \epsilon_\theta) \tag{18}$$

$$= \frac{2\bar{\alpha}_m}{1 - \bar{\alpha}_m}(\hat{x}_0' - \tilde{x}_0) \tag{19}$$

$$= \frac{2\bar{\alpha}_m}{1 - \bar{\alpha}_m}(\hat{x}_0' - \lambda\hat{x}_0 - (1 - \lambda)c_y) \tag{20}$$

$$= \left[ \frac{2\bar{\alpha}_m}{1 - \bar{\alpha}_m}(\hat{x}_0' - \hat{x}_0) \right] + \left[ \frac{2(1 - \lambda)\bar{\alpha}_m}{1 - \bar{\alpha}_m}(\hat{x}_0 - c_y) \right]. \tag{21}$$

**Gradients of $\mathcal{L}_{multi}$ Regarding $\hat{x}_0^{'}$.**

Similar to $\mathcal{L}_{GenDD}$, the gradient of $\mathcal{L}_{noise}$ regrading $\hat{x}_0^{'}$ is:

$$\nabla_{\hat{x}_0^{'}} \mathcal{L}_{noise} = \frac{2\bar{\alpha}_m}{1 - \bar{\alpha}_m}(\hat{x}_0^{'} - \hat{x}_0) \tag{22}$$

For the $\mathcal{L}_{CE}$ item, we assume the *weight* of the frozen linear classifier $\mathcal{C}_t(\cdot)$ is $W \in \mathbb{R}^{C \times d}$, where $d$ is the feature dimension and $C$ is the number of interested classes. Then,

$$\mathcal{L}_{CE} = -\log p(y|\hat{x}_0^{'}), \text{ where } p(y|\hat{x}_0^{'}) = \frac{W_y \hat{x}_0^{'}}{\sum_{i=1}^{C} W_i \hat{x}_0^{'}}, \tag{23}$$

$$\nabla_{\hat{x}_0^{'}} \mathcal{L}_{CE} = \sum_{i=1}^{C} \left[ p(i|\hat{x}_0^{'}) W_i \right] - W_y. \tag{24}$$

Overall, the gradients of $\mathcal{L}_{multi}$ regarding $\hat{x}_0$ is derived as:

$$\nabla_{\hat{x}_0^{'}} \mathcal{L}_{multi} = \gamma_0 \left[ \frac{2\bar{\alpha}_m}{1 - \bar{\alpha}_m}(\hat{x}_0^{'} - \hat{x}_0) \right] + \gamma_1 \left[ (\hat{x}_0^{''} - c_y) \right], \tag{25}$$

where $\hat{x}_0^{''} = \sum_{i=1}^{C} \left[ p(i|\hat{x}_0^{'}) W_i \right]$, and $c_y = W_y$. Note that we also use $c_y = W_y$ in **Definition** 1.

In Equation (25), we observe that $\hat{x}_0^{''} \approx \hat{x}_0^{'}$ when the predicted probability $p(y|\hat{x}_0^{'})$ approaches 1.0. To ensure consistency between training and inference in multi-task learning, it is crucial to employ multi-step sampling to obtain accurate estimates of $\hat{x}_0^{'}$ during the optimization of $\mathcal{L}_{CE}$. In such cases, the improved quality of $\hat{x}_0^{'}$ naturally leads to higher classification confidence, making the approximation $\hat{x}_0^{''} \approx \hat{x}_0^{'}$ practically valid.

**Summary.** Therefore, by comparing Equation (21) and Equation (25), we conclude that $\nabla_{\hat{x}_0^{'}} \mathcal{L}_{\text{GenDD}} \approx \nabla_{\hat{x}_0^{'}} \mathcal{L}_{\text{multi}}$. This indicates that GenDD, augmented with the distribution contraction mechanism, functions as a gradient-level surrogate for multi-task learning, without requiring the explicit application of the classifier loss to $\hat{x}_0^{'}$.

A.2 COMPARISONS OF SENSITIVITY TO HYPERPARAMETERS WITH PREVIOUS METHODS ON IMAGENET

Table 6: **Sensitivity to hyperparameters.** Advanced KD methods often involve complex hyperparameter tunning. Our *GenDD* method consistently works well across diverse teacher-student configurations with $\lambda = 0.9$ on ImageNet.

| Method | Teacher-Student | Hyperparameter configuration |
|---|---|---|
| KD | ResNet-34 — ResNet-18
ResNet-50 — MobileNet | $w_{kl} = 0.5, w_{ce} = 0.5, T = 1.0$ |
| DKD | ResNet-34 — ResNet-18
ResNet-50 — MobileNet | $w_{ce} = 1.0, w_\alpha = 1.0, w_\beta = 0.5, T = 1.0$
$w_{ce} = 1.0, w_\alpha = 1.0, w_\beta = 2.0, T = 1.0$ |
| IKL-KD | ResNet-34 — ResNet-18
ResNet-50 — MobileNet | $w_{ce} = 1.0, w_\alpha = 1.0, w_\beta = 0.5, T = 1.0$
$w_{ce} = 1.0, w_\alpha = 4.0, w_\beta = 1.0, T = 1.0$ |
| GenDD | ResNet-34 — ResNet-18
ResNet-50 — MobileNet | $\lambda{=}0.9$ |

## A.3 MORE RESULTS ON CIFAR-100

Table 7: **Top-1 accuracy (%) on the CIFAR-100 validation.** Teachers and students are in **different** architectures. All results are the average over 3 trials.

| Distillation Manner | | ResNet32×4 79.42 | WRN-40-2 75.61 | VGG13 74.64 | WRN-40-2 75.61 | ResNet32×4 79.42 |
|---|---|---|---|---|---|---|
| | Teacher | ShuffleNet-V1 | ShuffleNet-V1 | ResNet20 | ResNet8x4 | ShuffleNet-V2 |
| | Student | 70.50 | 70.50 | 69.06 | 72.50 | 71.82 |
| Discriminative Point-wise Distillation | FitNet | 73.59 | 73.73 | 69.27 | 74.74 | 73.54 |
| | RKD | 72.28 | 72.21 | 69.83 | 72.43 | 73.21 |
| | CRD | 75.11 | 76.05 | 71.16 | 76.64 | 75.65 |
| | OFD | 75.98 | 75.85 | - | 74.50 | 76.82 |
| | ReviewKD | 77.45 | 77.14 | - | 75.48 | 77.78 |
| | DKD | 76.45 | 76.70 | 70.57 | 75.23 | 77.07 |
| | IKL-KD | 76.64 | **77.19** | 70.88 | 76.12 | 77.16 |
| | KD | 74.07 | 74.83 | 70.20 | 74.01 | 74.45 |
| Gen.D.D. | **GenDD** | **77.58** | 76.73 | **72.17** | **77.55** | **78.13** |
| | Δ | **+3.51** | **+1.90** | **+1.97** | **+3.54** | **+3.68** |

## A.4 MORE ABLATIONS

**Ablation on MLP Head.** The diffusion head consists of a 3-layer MLP. To verify that the performance gains are not due to additional parameters, we insert a 2/3-layer MLP before the classifier in the student model for KD training. As shown in Table 8, this modification makes KD training even harder to optimize without improving results, indicating that the gains arise from the diffusion mechanism rather than extra parameters.

| Method | Top-1 (%) |
|---|---|
| KD*(optimal weight) | 75.61 |
| KD w/2-layer MLP | 75.02 |
| KD w/3-layer MLP | 74.60 |

Table 8: **Ablation on MLP head for the KD baseline.**

## A.5 DETAILED PERFORMANCE ON FEW-, MEDIUM-, AND MANY-SHOT

Table 9: **Top-1 accuracy(%) on the ImageNet-LT validation with GenDD.** "*" represents the unsupervised setting.

| Method | Teacher | Student | All | Few | Medium | Many |
|---|---|---|---|---|---|---|
| Baseline | - | ResNet-18 | 63.16 | 33.47 | 5.88 | 41.15 |
| Baseline | - | ResNet-50 | 67.25 | 38.56 | 8.21 | 45.47 |
| KD | ResNeXt-101 | ResNet-18 | 64.60 | 37.88 | 9.53 | 44.32 |
| KD | ResNeXt-101 | ResNet-50 | 68.83 | 42.31 | 11.37 | 48.31 |
| IKL-KD | ResNeXt-101 | ResNet-18 | 66.60 | 38.53 | 8.19 | 45.21 |
| IKL-KD | ResNeXt-101 | ResNet-50 | 70.06 | 43.47 | 10.99 | 49.29 |
| **GenDD*** | ResNext-101 | ResNet-18 | 66.71 | 39.02 | 8.66 | 45.54 |
| **GenDD*** | ResNeXt-101 | ResNet-50 | 70.12 | 43.52 | 10.84 | 49.31 |

## B LLM USAGE AND REPRODUCIBILITY

We only use ChatGPT to polish our writing.

Our code is available at

```
https://drive.google.com/file/d/12bwEj-wUqy2LFGpwsc_bikZF-Y_5SaP8/
view?usp=sharing
```

