# OpenReview forum: "Generative Distribution Distillation"
_ICLR.cc/2026/Conference — ICLR 2026 Conference Withdrawn Submission_

### Official Review · Reviewer_vNZk · 2025-10-16

**Soundness:** 2
**Presentation:** 2
**Contribution:** 2
**Rating:** 2
**Confidence:** 4

**Summary:**

This paper proposed GenDD, a feature-based knowledge distillation method mainly for classification models. GenDD proposed to train a conditional diffusion model with the student's feature as the condition and with the teacher's feature as the generated output. For better class-awareness, GenDD linearly scales the teacher's feature of each sample towards its class centroid.

**Strengths:**

1. The experiments cover a wide range of models and datasets.
2. The writing is clear and easy to follow.

**Weaknesses:**

1. The proof of Theorem 1 is questionable. A critical step of this proof is line 776, where the authors claim $W _y = c _y$. However, this claim is not close to the truth without any other assumptions, and the authors did **not** mention this assumption elsewhere in the main body of the paper. If this problem is not addressed, then the proof can be viewed as invalid.
2. The motivation of the proposed method is unclear and not well justified. Since the features of both teacher and student models are deterministic, it is not obvious why a generative model (such as a diffusion model) is used to predict the teacher's feature. In other words, it is not clear why it is beneficial to do "distributional KD". A simple deterministic neural network is conceptually more favorable to parameterize this mapping, as the baselines did, because it is much easier to train and does not have the non-differentiable problem GenDD has.
3. Apart from an abbreviation, the authors did not provide any citation or description for each baseline. This makes it difficult to assess the empirical rigor.
4. The related works section is currently a placeholder. For the KD paragraph, the authors only provided a condensed citation block. The authors did not sufficiently identify how their method is different from existing literature or provide any analysis on conceptually related designs.
5. According to the ablation study, the method seems to be highly sensitive to hyperparameter choice, especially lambda.

**Questions:**

Please see the weaknesses above.

---

### Official Review · Reviewer_Phrj · 2025-10-22

**Soundness:** 1
**Presentation:** 3
**Contribution:** 2
**Rating:** 2
**Confidence:** 5

**Summary:**

Instead of proposing to just match logits or features, the authors use the student features to condition a diffusion model to generate the teacher features. At test time they then reconstruct the teacher features and classify using the teachers classifier. To address the problem of high dimensional optimisation and improve training stability, they introduce a SplitTok as a different way to condition the diffusion model. Finally, they naturally introduce label supervision as a multi-task problem.

The authors provide experiments on small and large scale datasets.

**Strengths:**

The method is novel and quite interesting to read. In contrast to prior works, which just introduce an additional alignment loss, they propose to model the problem as a generative task, where the reconstruction loss can be seen as a regularization to align with the teacher. This incorporation of label supervision is very natural.

There is a good motivation for the SplitTok component and it is clearly ablated to show its importance.

The ablation experiments are on ImageNet, which is really good to see. Ablations on specific architecture pairs on CIFAR can be misleading.

**Weaknesses:**

In abstract: "With label supervision, our ResNet-50 achieves 82.28% top-1 accuracy on ImageNet in 600 epochs of training". This statement is misleading. [1] is a very well known KD paper that also explicitly does label supervision for training a ResNet-50 model and achieves an accuracy is 82.8%. If the authors wish to highlight this result, they should say state-of-the-art within this given training budget.

Missing related work [2, 3, 4], which to my understanding do not have any extensive hyper-parameter tuning. In fact they all show robustness to hyperparameters empirically.
Furthermore, [4, 5] show better results across all the CIFAR architecture pairs presented. Although the CIFAR benchmark is admittedly quite saturated, omitting a comparison to many of the recent KD methods (especially those for *feature/representation* distillation) is very misleading. It seems the authors only compare against feature methods that are over 4 years old.

The practical motivation for this work is unclear. Conventionally distillation is used to train smaller models so that they are faster at inference. In this work, they propose a reconstruction based task which incurs inference overheads and additionally use the teachers classifier. Firstly, there is already work suggesting the use of sharing the teachers classifier [6] but they introduced pruning so that this modification did not incur any parameter overhead for a fair comparison to other KD works. Secondly, there are no timing results for the reconstruction during inference and no attempts to make the evaluation match the evaluation cost with other KD methods they compare to. In practice, if there is no interest in the evaluation overhead, how would SplitTok compare to just using a larger student model?

The way I see it is that this proposed method is theoretically very interesting, but practically not too useful. I would encourage the authors to find a useful practical utility for this method over prior KD works. As it stands, comparing to cheaper (at evaluation time) KD methods is misleading, and claiming that prior works are sensitive to hyperparameters is not true (not just the sensitivity of vanilla logit distillation with the temperature parameter). Related to this point, in figure 4b there is only a noticeable improvement over the baseline KL when 1-\lambda = 0.9. I'm not sure how this highlights robustness to \lambda? in fact it shows that a very specific value of lambda is needed to improve upon the baseline KL (71.4% mobilenet).

[1] Knowledge distillation: A good teacher is patient and consistent. CVPR 2022

[2] Understanding the Role of the Projector in Knowledge Distillation. AAAI 2024

[3] Wasserstein Contrastive Representation Distillation. CVPR 2021

[4] Logit Standardization in Knowledge Distillation. CVPR 2024

[5] Multi-level logit distillation. CVPR 2023

[6] Knowledge Distillation with the Reused Teacher Classifier. CVPR 2022

Minor comments:

L086: "Conditoned"

**Questions:**

See weaknesses

---

### Official Review · Reviewer_VbvG · 2025-10-28

**Soundness:** 3
**Presentation:** 3
**Contribution:** 3
**Rating:** 6
**Confidence:** 5

**Summary:**

This paper recasts the point-wise discriminative paradigm of knowledge distillation into a diffusion-based generative one. It introduces two techniques, namely Split Tokenization and Distribution Contraction to improve performance. Experimental results on different datasets, models, and settings demonstrate the effectiveness of the proposed method.

**Strengths:**

1.Casting the point-wise discriminative knowledge distillation paradigm into a generative one is conceptually novel and well-motivated.

2.Experimental results show that the proposed method is effective and yields significant gains on different datasets and settings.

**Weaknesses:**

1.Relevant (e.g., MGD [1]) or recent strong KD baselines (e.g., LSKD [2], CRLD [3]) are missing for introduction and comparison. For example, Tables 2 and 5 only present outdated KD methods, and recent strong baselines such as FCFD [4], LSKD [2], CRLD [3], and SDD [5], are absent for comparison.

2.The diffusion process is notorious for its time-consuming multi-step sampling. The 1000-step and 64-step sampling used in the training and inference of the proposed method could make it substantially inefficient during both training and deployment. Meanwhile, detailed benchmarking and discussion of this aspect are not provided.

3.Choices for some diffusion-related hyperparameters are not justified. For example, how is the CFG guidance scale of 2.0 chosen? Why 64-step for inference? What are the effect of using different values?

4.Can the proposed method be applied to tasks beyond classification, for example object detection? The practical value of the proposed method may be limited if it only works for classification.

[1] Yang et al. Masked Generative Distillation. ECCV 2022.

[2] Sun et al. Logit Standardization in Knowledge Distillation. CVPR 2024.

[3] Zhang et al. Cross-View Consistency Regularisation for Knowledge Distillation. ACM MM 2024.

[4] Liu et al. Function-Consistent Feature Distillation. ICLR 2023.

[5] Wei et al. Scale Decoupled Distillation. CVPR 2024.

**Questions:**

Please see Weaknesses.

---

### Author Response · Authors · 2025-11-13
**Clarification on questions**

We thank the reviewers for their valuable comments. Here, we clarify some misunderstandings about our work.


1. Sensitivity to hyperparameters.

    (1) Traditional KD with KL loss has at least 2 hyperparameters: the temperature $\tau$ and the weight for KL.
         Different teacher-student configurations can have different optimal weight for KL loss as shown in Figure 2.
         Different $\tau$ could be employed, regularly 1.0 on ImageNet, 4.0 on CIFAR.

    (2) For advanced KD methods, like DKD and IKL-KD, there are often other hyperparameters. Optimal hyperparameters can also be different for different teacher-student configurations. We show it in Appendix A.2.

    (3) The $\lambda$ controls the degree we use label supervision in GenDD. It can have great effects on model performance.
          However, we use $\lambda=0.9$ on ImageNet for all of the teacher-student configurations, demonstrating the robustness of GenDD.




2. Training settings for the new state-of-the-art on ImageNet with ResNet-50.

    We evaluate our model on two settings: (1) regular training in 100 epochs with RandomResizedCrop and horizontal flip, (2) strong training settings A1 and A2 that are defined in our paper.

   Our state-of-the-art performance is obtained under the A1 strategy.

3. Addtional parameters for GenDD.

    Our paper presents an alternative generative pipeline for knowledge transfer. Especially, it can also do unsupervised KD effectively.
    The diffusion head can have extra parameters. We have conducted ablations to show that the performance gains come from the diffusion mechanism rather than extra parameters in Table 8.

---

### Note · Authors · 2025-11-13

**Comment:**

Thanks for the valuable comments from the reviewers.

**Withdrawal Confirmation:**

I have read and agree with the venue's withdrawal policy on behalf of myself and my co-authors.